# Low-noise encoding of active touch by layer 4 in the somatosensory cortex

**Samuel Andrew Hires[†‡], Diego A Gutnisky[†], Jianing Yu, Daniel H O'Connor[§], Karel Svoboda***

Janelia Research Campus, Howard Hughes Medical Institute, Ashburn, United States

**Abstract** Cortical spike trains often appear noisy, with the timing and number of spikes varying across repetitions of stimuli. Spiking variability can arise from internal (behavioral state, unreliable neurons, or chaotic dynamics in neural circuits) and external (uncontrolled behavior or sensory stimuli) sources. The amount of irreducible internal noise in spike trains, an important constraint on models of cortical networks, has been difficult to estimate, since behavior and brain state must be precisely controlled or tracked. We recorded from excitatory barrel cortex neurons in layer 4 during active behavior, where mice control tactile input through learned whisker movements. Touch was the dominant sensorimotor feature, with >70% spikes occurring in millisecond timescale epochs after touch onset. The variance of touch responses was smaller than expected from Poisson processes, often reaching the theoretical minimum. Layer 4 spike trains thus reflect the millisecond-timescale structure of tactile input with little noise.

**\*For correspondence:**
svobodak@janelia.hhmi.org

[†]These authors contributed equally to this work

**Present address:** [‡]University of Southern California, Los Angeles, United States; [§]Johns Hopkins University School of Medicine, Baltimore, United States

**Competing interests:** The authors declare that no competing interests exist.

## Introduction

Variability in spike trains constrains how neural computations can be implemented (*London et al., 2010*; *Renart and Machens, 2014*). Measured cortical spike trains are often irregular in time, and spike counts vary over repeated presentations of identical sensory stimuli (*Tolhurst et al., 1983*; *Shadlen and Newsome, 1998*; *Maimon and Assad, 2009*). One view holds that this variability is irreducible and therefore represents noise (*Shadlen and Newsome, 1998*; *London et al., 2010*). Noisy spike trains are difficult to reconcile with the integrative properties of single neurons (*Softky and Koch, 1993*) and the high reliability of cortical neurons (*Mainen and Sejnowski, 1995*) and synapses (*Stevens and Zador, 1998*). This discrepancy has motivated models of cortical circuits that inherently produce noisy spike trains even with reliable neurons, by virtue of chaotic dynamics (*van Vreeswijk and Sompolinsky, 1996*; *Shadlen and Newsome, 1998*; *Litwin-Kumar and Doiron, 2012*). Irregular spike trains suggest that spike rates, but not spike timing, are used by the brain for computation (*Mazurek and Shadlen, 2002*; *London et al., 2010*).

Another view contends that variability in cortical spike trains is not noise but reflects fluctuating hidden states with possible behavioral significance and uncontrolled experimental factors (*Gur et al., 1997*; *Kara et al., 2000*; *DeWeese et al., 2003*; *VanRullen et al., 2005*; *Amarasingham et al., 2006*). Measuring and accounting for these hidden states may reveal the detailed structure of spike trains to be deterministic and predictable. For example, minimizing fixational eye movements in alert monkeys reduces spike count variability in the visual cortex (*Gur et al., 1997*). In the sensory periphery, spikes are often coupled to stimulus features with high temporal precision. This precision allows timing-based neural codes to be faster (*Johansson and Birznieks, 2004*) and more efficient (*Gollisch and Meister, 2008*) than rate-based codes.

Multiple factors can shape neural spike trains, only a subset of which are controlled or measured in typical experiments. Uncontrolled factors will add to measured variability (*Masquelier, 2013*; *Renart and Machens, 2014*). These factors can be external to the brain, such as sensory stimuli

**eLife digest** Cells called neurons connect to form large networks that process information in the brain. A region of the brain called the cerebral cortex receives information about touch from sensors in the skin. A series of neurons relay the touch information to the cerebral cortex as patterns of electrical activity called 'spike trains'. Understanding how these spike trains represent information about the world around us is one of the greatest challenges facing neuroscience.

At first glance, the number and timing of the individual spikes within the trains appear to be random. It is possible that the irregularity within spike trains is 'noise' that is generated within the cortex itself. This noise could represent uncertainty about the nature of the stimulus from the sensors, or random fluctuations in brain activity. However, other findings have challenged this view and argued that these erratic spike trains actually carry hidden information.

Hires et al. investigated this possibility by recording how neurons within a region of the mouse brain called the somatosensory cortex responded to sensory information coming from the mouse's whiskers. Mice sweep their whiskers across objects to locate and identify them, much like how humans feel objects with their fingertips. Here, the mice used their whiskers to judge the location of an object by touch alone, while the electrical activity of the neurons was measured using electrodes. Importantly, the movements of the whiskers and contact with the object were tracked to one millisecond precision.

Similar to previous studies, sensory information from the whiskers triggered irregular spike trains in neurons within the somatosensory cortex. Hires et al. found that the apparently irregular spikes coincided precisely with the timing of when the whiskers contacted the object. Other spikes aligned perfectly with the movement of whiskers into particular positions. Furthermore, the patterns of electrical activity in the spike trains precisely predicted when and how the object was contacted, and which whisker was involved.

These findings suggest that the timing of individual spikes within spike trains carries important information to the brain. Future studies will develop our understanding of how the brain interprets and responds to the rich data contained in these spike trains to identify objects and decide how to interact with them.

(*Baudot et al., 2013*), or internal to it, such as behavioral state (*Mitchell et al., 2007*, *2009*; *Churchland et al., 2010*). They can be fundamentally irreducible, such as channel noise and chaotic dynamics of neural networks (*van Vreeswijk and Sompolinsky, 1996*), or potentially controllable, such as animal behavior (*Gur et al., 1997*) and fluctuating input from other brain areas (*Gomez et al., 2013*). Here we assessed the precision of spikes during active tactile behavior. We recorded from neurons in layer 4 (L4) in the mouse barrel cortex and measured tactile behaviors with high temporal and spatial resolution.

L4 contains a precise map, where individual barrels process information from single whiskers (*Simons, 1978*). Similar to other cortical circuits, connectivity within L4 is highly recurrent (*Lefort et al., 2009*). However, the only major long-range input into L4 ascends from sensory neurons via the posterior medial thalamus into L4 (*Lu and Lin, 1993*; *Bureau et al., 2006*; *Hooks et al., 2011*). L4 thus receives little uncontrolled input.

During natural behavior animals move their sensors to acquire information about the world (*Deschenes et al., 2012*). Measurement of spiking statistics in the barrel cortex is thus most meaningful during active sensation, when mice shape sensory input by moving their whiskers to solve a tactile task. In our experiments mice localized objects with their whiskers (*Knutsen et al., 2006*; *O'Connor et al., 2010a*; *O'Connor et al., 2013*). Activity within the barrel cortex is necessary for whisker-based sensation (*Hutson and Masterton, 1986*; *O'Connor et al., 2010*; *Guo et al., 2014*). Whisker deflections (*Simons, 1978*; *Bruno and Sakmann, 2006*; *Jadhav et al., 2009*) or active touch (*Crochet and Petersen, 2006*; *Curtis and Kleinfeld, 2009*; *O'Connor et al., 2010b*; *O'Connor et al., 2013*) trigger temporally sharp responses in barrel cortex neurons, which underlie the perception of object location (*Diamond et al., 2008*; *O'Connor et al., 2013*). Spike rates of barrel cortex are also modulated by whisker movements on multiple time scales (*Fee et al., 1997*; *Crochet and Petersen, 2006*; *Curtis and Kleinfeld, 2009*). Similar to visual cortical neurons (*Tolhurst et al., 1983*;

*Shadlen and Newsome, 1998*; *Maimon and Assad, 2009*), barrel cortex neurons respond with high trial-to-trial variability to passive sensory stimulation (*Wang et al., 2010*; *Adibi et al., 2013*).

To minimize uncontrolled variability we thus recorded cortical spike trains in a well-characterized neural circuit, in mice engaged in active sensorimotor behavior, with precisely quantified sensory input. We show that spikes in L4 are temporally precise and have spike count variance close to the theoretical minimum. This precision allows efficient decoding of touch timing from small numbers of L4 neurons, supporting a role for temporal coding in cortical computation.

## Results

### L4 responses are temporally precise

We trained mice to locate an object by active touch with a single whisker (C2) (*O'Connor et al., 2010a*; *O'Connor et al., 2013*) (n = 21 mice, 52 sessions; fraction of trials correct, 0.740 ± 0.086; mean ± s.d.; *Figure 1A*; *Figure 1—figure supplement 1*). Single whisker experiments allowed us to track the relevant tactile variables with high precision during behavior. In each trial, during a sample epoch lasting a few seconds (1.54–4.05, mean 2.39 s), a pole appeared in one of two locations on the right side of the head. High-speed videography and automated whisker tracking quantified whisker movement (azimuthal angle, θ; whisking phase, ϕ), changes in curvature caused by the forces exerted by the pole on the whisker (change in curvature, Δκ) (*Birdwell et al., 2007*; *Pammer et al., 2013*), and contact time, all with 1 millisecond temporal precision (*Clack et al., 2012*; *O'Connor et al., 2013*) (*Figure 1A,B*). Mice whisked in bouts (mean bout duration, 261 ms; peak-to-peak amplitude, 15.7°; frequency, 15.4 Hz) interspersed with periods of rest. Mice touched the pole multiple times (mean number of touches, 2.33) before reporting perceived object location with licking (mean reaction time 367 ± 234 ms; mean ± s.d.) (*Figure 1—figure supplement 1*).

We targeted recordings to excitatory neurons in L4 of the principal barrel corresponding to the spared whisker (31 cells in C2; 10 additional outside C2) and L5 near the principal barrel (11 cells), guided by intrinsic signal imaging (number of trials per neuron 115 ± 60; *Figure 1—figure supplement 2*). We recorded single units via loose-seal juxtacellular methods to avoid potential artifacts of intracellular disruption from whole-cell patch or misassigned spikes from spike sorting (*DeWeese et al., 2003*). The dynamics of a typical neuron in L4 C2 is illustrated in *Figure 1*. The neuron had a low baseline firing (0.73 spk/s; 500 ms before start of sample epoch) that increased substantially during the sample epoch (2.6 fold; to 1.91 spk/s; p = 3.38e-7, Wilcoxon rank sum) (*Figure 1C,D*).

Neuronal variability can arise from external factors, such as trial-to-trial variations in behavior, or internal factors, such as synaptic noise and fluctuating motivation and arousal (*Renart and Machens, 2014*). The Fano factor (FF) is a widely used measure of variability in spike trains (*Berry et al., 1997*). FF is defined as the variance of the spike count divided by the mean spike count over some time window. For a Poisson process, FF = 1, independent of the window size. In our task, mice are free to explore the object differently in each trial. At a coarse scale, behavior and neural responses were irregular during object localization. FF computed by counting spikes over the entire sample epoch (stimulus presentation), was huge (FF = 7.51). Since each trial corresponds to different whisker movements and different patterns of touches this value of FF includes extrinsic variability due to behavior, in addition to intrinsic variability. Aligning spikes to the fine-scale structure of behavior revealed that spikes were mainly coupled to temporally irregular sensory input from object contact (*Figure 1E*). Spike rate was sharply elevated shortly after touch onset (*Figure 1F*). Each touch evoked on the order of one spike (1.51 spikes/first touch; 0.31/later touches) with short latency (onset, 8 ms) (*Table 1*; *Figure 2—figure supplement 1*). The large FF when computing spikes over the sample period is therefore at least in part due to trial-to-trial variability in active touch.

Spike times outside of touch periods also appeared temporally irregular. Firing rates were low in both whisking (0.67 spikes/s) and non-whisking (0.23 spikes/s) periods. Yet during active exploration, the timing of spikes was coupled to the phase of rhythmic whisker movement (*Fee et al., 1997*; *Curtis and Kleinfeld, 2009*), with a modulation depth close to 1 (*Figure 1G,H*; *Figure 2—figure supplement 1*). For the example neuron, using whisker behavior it is possible to predict time windows where a neuron will fire a single or small number of spikes, as well as time periods when the spike probability is zero (*Figure 1G,H*; phase, 0), similar to spike trains measured in the salamander retina (*Keat et al., 2001*). In this sense the timing of each spike encodes a tactile feature (touch and whisking phase).

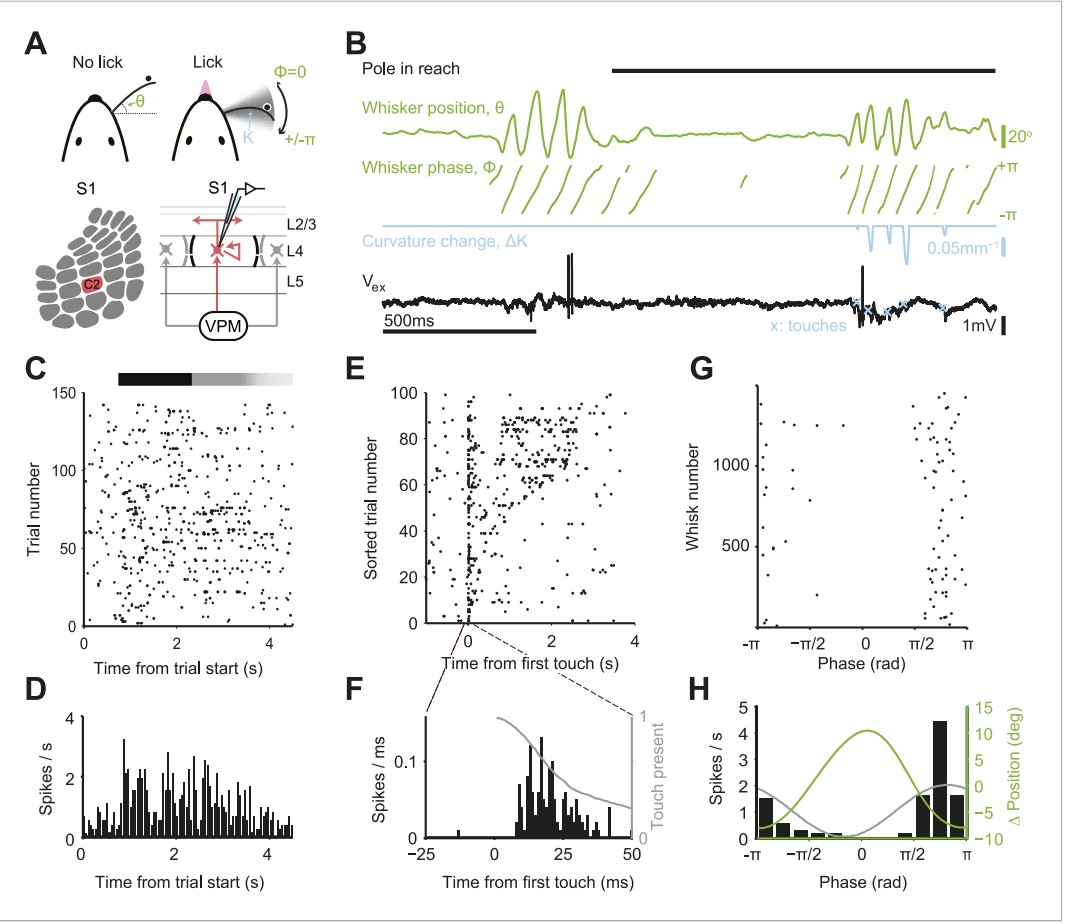

**Figure 1**. Activity during tactile behavior in a layer 4 excitatory cell. (**A**) Top, mice judged object location with a single whisker. Whisker position (azimuthal angle, $\theta$), whisking phase ($\phi$), and whisker curvature ($\kappa$) were measured from video recordings. Bottom, recordings were made from excitatory cells in the principal barrel (red). L4 excitatory neurons receive excitatory input from VPM and excite each other within individual barrels. (**B**) Behavioral and electrophysiological data (single trial). $\theta$, whisker position (green); $\phi$, whisking phase (green); $\Delta\kappa$, change in whisker curvature (blue), which is proportional to pressure on mechanoreceptors at the base of the whisker; $V_{ex}$, extracellular spike waveform (black) recorded in loose-seal mode (blue crosses, touch onsets). The black horizontal bar indicates the time when the object was in reach. (**C**) Spike raster for one example neuron. Same data as in **D–H**. Pole in reach for all trials (black bar) with variable exit time (grey bar). (**D**) Peri-stimulus time histogram aligned to the trial onset (bin size, 50 ms). (**E**) Spike raster aligned to first touch, and sorted according to last touch in the sample period (late on top). Trials without touch are not shown. (**F**) Peri-stimulus time histogram aligned to first touch (bin size, 1 ms). The grey line represents the proportion of touches with durations >= than time (max of 1). (**G**) Spikes aligned by whisking phase in a whisking bout (whisking amplitude >2.5° peak-to-peak). Only exploration periods excluding touch were used. (**H**) Spike histogram aligned to whisking phase (bin size, 30°) Best-fit spike modulation (grey). Average change in whisker position/bout (green).

The following figure supplements are available for figure 1:

**Figure supplement 1**. Behavior during cell-attached recordings.

**Figure supplement 2**. Targeting recordings.

---

The temporally precise spiking after touch was restricted to L4 neurons in the principal barrel. Neurons recorded in the C2 barrel column showed brief responses to touch (*Figure 2A–C*; *Figure 3—figure supplement 1*; *Table 1*). In L4, but outside of C2, touch responses were much weaker (first touch, p = 1.4e-5; later touches, p = 6.32e-6; Wilcoxon rank sum). Layer 5 neurons near C2 had much higher firing rates, with touch responses that were more diverse (*Figure 2B*;

**Table 1.** Spiking responses of recorded neurons

| Area | Spikes/touch first touch | Spikes/touch later touch | Spikes evoked (touch) | Spikes evoked (touch and whisking) | Phase modulation depth | Non-whisking spike rate (spk/s) | Whisking spike rate (spk/s) | Onset latency (ms) | Minimum ISI (ms) |
|---|---|---|---|---|---|---|---|---|---|
| L4 | 1.36 ± 1.32 | 0.74 ± 0.87 | 70.6 ± 20.9% | 75.2 ± 19.0% | 0.67 ± 0.32 | 1.31 ± 2.88 | 1.39 ± 2.33 | 7.8 ± 3.0 | 2.9 ± 1.9 |
| Inside C2 | 0.87 | 0.33 | 74.7% | 80.3% | 0.70 | 0.32 | 0.33 | 8 | 2.3 |
| (n=31) | (0.08–5.77) | (0.10–2.92) | (24.5–96.0%) | (35.3–96.8%) | (0–1.00) | (0.02–13.7) | (0.02–10.9) | (4–18) | (1.2–11.0) |
| L4 | 0.08 ± 0.19 | 0.02 ± 0.04 | 9.2 ± 13.7% | 33.0 ± 30.2% | 0.58 ± 0.36 | 2.20 ± 2.91 | 2.41 ± 3.48 | 18.5 ± 6.0 | 3.3 ± 1.3 |
| Outside C2 | 0 | 0 | 3.6% | 23.8% | 0.61 | 1.10 | 1.31 | 18 | 2.8 |
| (n=10) | (0–0.61) | (0–0.12) | (0–41.0%) | (1.0–79.4%) | (0.10–1.00) | (0.03–8.86) | (0.17–11.8) | (12–26) | (2.3–6.7) |
| L5 | 1.95 ± 2.96 | 1.28 ± 1.88 | 24.0–18.6% | 31.5 ± 16.4% | 0.18 ± 0.12 | 12.0 ± 9.87 | 16.2 ± 15.3 | 9.7 ± 5.2 | 3.7 ± 2.5 |
| Near C2 | 0.98 | 0.42 | 24.4% | 36.0% | 0.13 | 12.8 | 13.5 | 8 | 2.7 |
| (n=11) | (0–8.68) | (0–5.47) | (0.5–55.5%) | (1.4–56.1%) | (0.06–0.46) | (0.23–29.7) | (1.00–54.0) | (4–20) | (1.6–9.1) |

Mean ± standard deviation; median; (range).

*Figure 3—figure supplement 2*; *Table 1*). Modulation by whisking phase was not significantly different between L4 neurons inside and outside the C2 barrel (p = 0.68), but both were significantly more phase modulated than L5 (p = 4.3e-5 and p = 0.012 respectively, Wilcoxon rank sum) (*Figure 2—figure supplement 1*). There were no significant changes in firing rate between whisking and non-whisking across the L4 population (L4 inside C2, p = 0.75; L4 outside C2, p = 0.92), whereas L5 showed a modest, but significant increase with whisking (p = 0.019, Wilcoxon signed rank).

We estimated the proportion of spikes that were temporally coupled to sensory input for each L4 neuron during whisking or touch in the sample epoch (exploration time). We counted the fraction of spikes falling into a small time window after touch (*Figure 3A,B*). This proportion increases with expanding window size, resulting in an initially steep curve (*Figure 3C*). At some window size, the curve levels off as the proportion of touch spikes increases no faster than expected for a random spike train at the neuron's mean spike rate. This transition point defines the proportion of touch-coupled spikes. For the neuron illustrated in *Figure 3A*, 74.4% spikes fall into a time window spanning 8 to 40 ms after touch onset, comprising 15.3% of overall exploration time. This implies temporally sparse spiking (*Berry et al., 1997*). Over the L4 C2 population, 70.6% ± 20.9% (mean, s.d.) of spikes were coupled to touch in an average of 14.9% of exploration time. L4 neurons outside of C2 had significantly less touch-evoked spikes, 8.4% ± 13.3% in 3.5% of exploration time (p = 6.53e-6, Wilcoxon rank sum). L5 neurons in C2 also showed significantly less touch-evoked spikes, 24.0% ± 18.6% touch spikes in 19.0% of exploration time (p = 2.30e-5, Wilcoxon rank sum) (*Figure 3C*).

Using a similar approach we measured the proportion of the remaining spikes coupled to the phase of whisker movement (*Figure 3A,B,D*). Overall, for neurons in C2 75.2 ± 19.0% of spikes were coupled to touch or whisking phase in an average of 22.0% of exploration time, significantly more than outside of C2 (33.0 ± 30.2 in 16.9% of time; p = 2.88e-4), or in L5 (31.5 ± 16.4 in 26.1% of time; p = 2.96e-5, Wilcoxon rank sum). Subtraction of the touch spikes from touch and phase spikes reveals that 4.6 ± 8.7% of spikes were purely phase coupled in L4 C2, vs 23.8 ± 27.9% outside of C2 and 7.6 ± 14.9% in L5. We conclude that during object localization, the majority of spikes in L4 excitatory neurons encode aspects of touch of the primary whisker, with many of the remaining spikes encoding whisking phase.

## Decoding touch

The temporal precision of spiking in L4 could be used to extract behaviorally relevant information. We assessed the ability of simulated populations of L4 neurons to detect touch and discriminate touch timing and whisking phase.

We used a simple model based on resampling the recorded spike trains measured in L4 neurons ('Materials and methods'). Pooling activity from only fifteen L4 neurons (out of approximately 1600

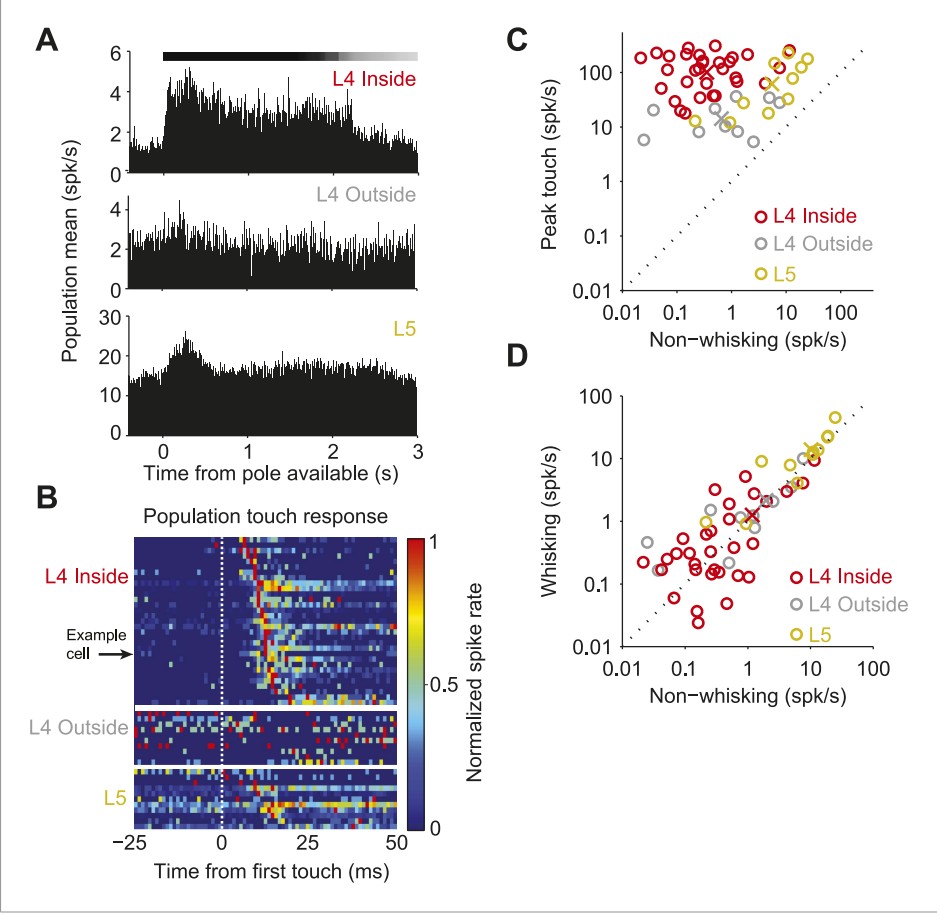

**Figure 2**. Neural responses to behavioral variables across three populations. (**A**) Grand mean peri-stimulus time histogram. Top, L4 in the C2 barrel (31 neurons); middle, L4 outside of C2 (10 neurons); bottom, in L5 near C2 (11 neurons). Pole in reach for all trials (black bar) and some trials (grey bar). (**B**) Peri-stimulus time histograms aligned to first touch. Top, neurons in L4 C2 sorted by time of peak touch response; middle, L4 neurons outside of C2; bottom, neurons in L5 near C2. Arrow head points to the same neuron as *Figure 1B–H*. (**C**) Peak spike rate after touch (1 ms bin) vs spike rate in the absence of whisking, individual cells (o), population means (x). Red, L4 inside C2; grey, L4 outside C2; yellow L5 near C2. (**D**) Spike rate during whisking compared to spike rate in the absence of whisking. Symbols as in **C**.

The following figure supplement is available for figure 2:

**Figure supplement 1**. Some population characteristics of all recordings.

[*Lefort et al., 2009*]) in C2 was sufficient to detect 95% of touches (integration time, 10 ms) (*Figure 4A*). Touch detection by neurons from surrounding barrels (>200 μm from the principal whisker) was poor (*Figure 4A*). Beyond detection, a group of 200 neurons in C2, integrating in 10 ms windows after touch, also allowed decoding of elapsed time from touch with high precision (minimum 0.55 ms uncertainty with 95% confidence at 10 ms post-touch onset using a naïve Bayes decoder [*Duda et al., 2001*]) (*Figure 4B*; *Figure 4—figure supplement 1*). This implies that a decoder reading L4 activity can determine which whisker makes contact with millisecond temporal precision (*Panzeri et al., 2014*).

In contrast, a large population of neurons (1000) was required to provide even a rough estimate of whisking phase (24° in C2, 33° outside at d′ = 1 performance, equivalent to 76% correct discrimination) in C2 and the surround columns (*Figure 4C*). The poor performance of the decoder is related to the low spike rate during whisker movements. Thus, the timing of spikes in L4 barrel cortex provides only a coarse representation of whisking phase.

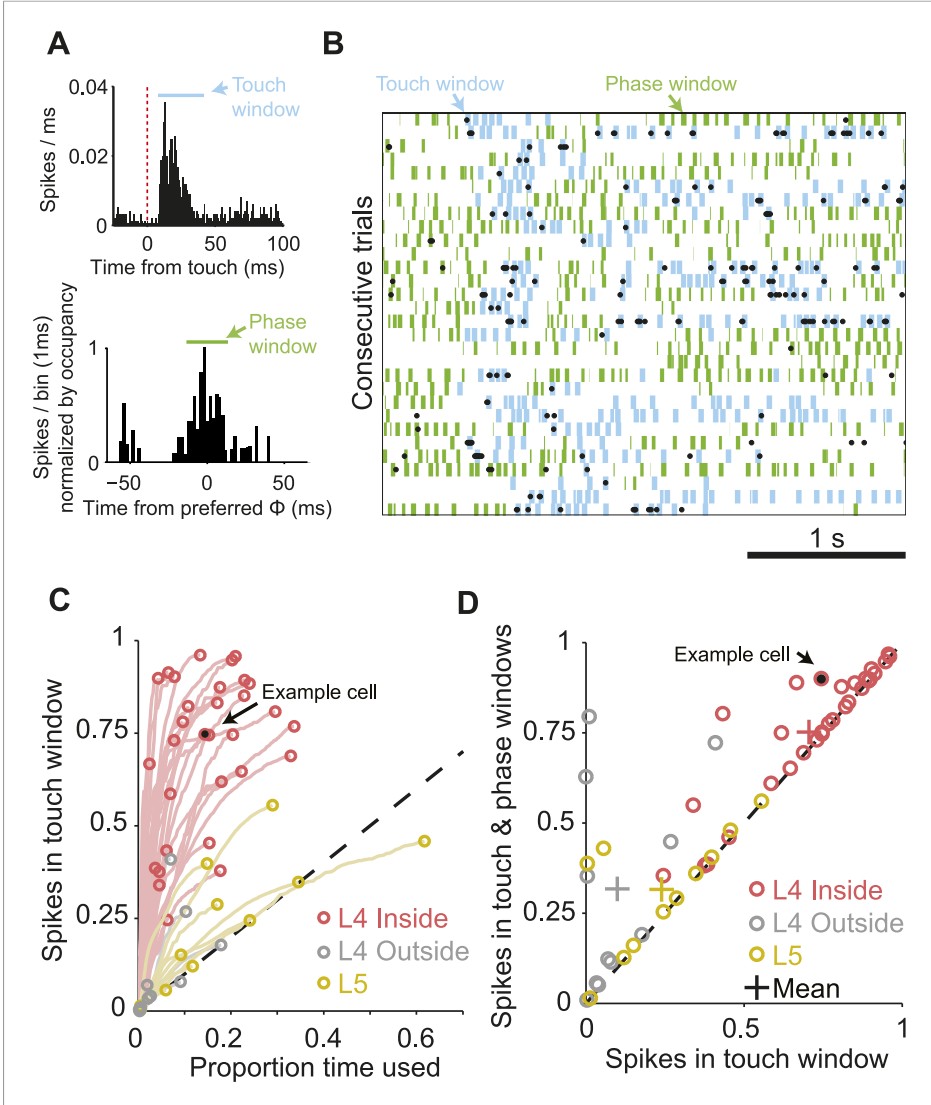

**Figure 3**. The majority of spikes in L4 excitatory neurons encode tactile information. (**A**) Top, peri-stimulus time histogram of example cell from *Figure 1* for all session touches with overlaid touch window (8–40 ms) (blue). Bottom, peri-stimulus spike histogram (PSTH) aligned to preferred whisking phase with overlaid window (green) aligned with the peak of the PSTH. Bins are normalized by occupancy. (**B**) Raster plot aligned to the sample period. Overlays: Blue, touch windows from **A**. Green, whisking phase windows from **A**, centered on the maximum in the phase-aligned spike histogram. Phase windows can be truncated at the margins of whisk cycles resulting in variable window lengths. (**C**) Proportion of spikes in touch window as a function of time in trial. Red, L4 inside C2; grey, L4 outside C2; yellow L5 near C2. Lines show the evolution of touch spikes as touch windows expand from the onset latency of each cell. Circles indicate the proportion of spikes coupled to touch at the final touch window size for each cell. (**D**) Proportion of spikes coupled to touch and whisking phase vs to touch alone. Colors as in **C**.

The following figure supplements are available for figure 3:

**Figure supplement 1**. Response characteristics of all L4 recordings in C2.

**Figure supplement 2**. Response characteristics of all L4 recordings outside of C2 and L5 recordings near C2.

## Low spike count variability

A hallmark of cortical spike trains is high variability in spike count over repeated presentations of identical stimuli (*Renart and Machens, 2014*). Variability in the number of spikes evoked reflects both

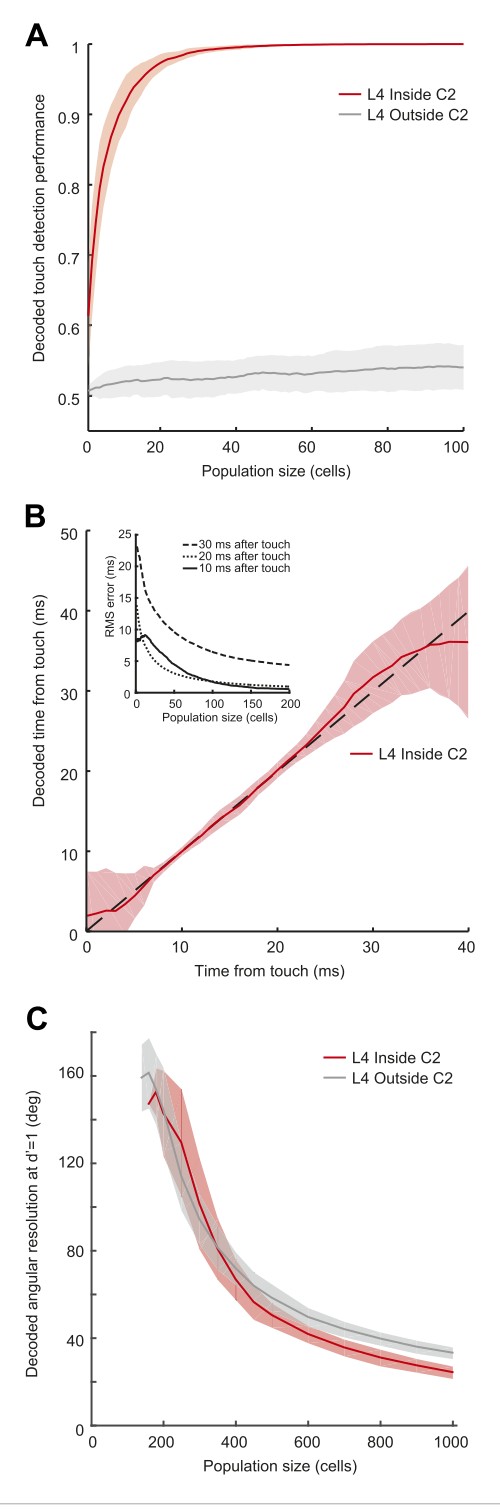

**Figure 4**. Decoding of touch and phase from L4 spikes. (**A**) Decoding of touch by a linear decoder of pooled activity. We randomly selected a variable number of neurons in two separate sets of neurons corresponding to L4 barrels inside C2 (red) and outside of C2 (gray). We pooled the activity of all the neurons in each of the sets and integrated the neural activity for 10 milliseconds.
*Figure 4. continued on next page*

variability in behavior (e.g., the number and quality of touches) and irreducible noise intrinsic to the cortical circuit. We thus analyzed variability aligned to individual touches while accounting for differences in the properties of individual touches.

In our active object localization task, mice produce tactile input through whisker movements, which varies greatly across trials and individual touches within a trial (*O'Connor et al., 2010*). To explore how sensory responses depended on different tactile stimulus features we sorted touches and the L4 neuron responses by one of three behavioral variables: the order of touch within each trial, to account for adaptation; whisker velocity just before touch, to account for rate of change of impact forces; maximum whisker curvature during touch, which is proportional to peak touch force (*Figure 5A,B*). Touch response magnitude was highly modulated by each of the three variables (mean modulation index: touch order, 0.71 ± 0.25; velocity, 0.71 ± 0.27; max curvature, 0.72 ± 0.23; mean ± std; *Figure 5A–C*). Responsiveness to pretouch velocity and max curvature covaried strongly (pairwise correlation coefficient 0.80, p = 4.3e-8), whereas touch order response was somewhat less correlated with velocity and curvature (correlation coefficients 0.70, 0.65, p = 1.0e-5, p = 8.2e-5). The deep modulation index of sensory responses to tactile stimulus features indicates that stimulus variability likely accounts for a significant component of variability in the touch response.

To put an upper bound on how much spike count variation in our recordings derives from irreducible noise intrinsic to the cortical circuit, it is critical to sort touch events by stimulus characteristics and compute the FF across touches with similar features. A density-based clustering algorithm (*Ester et al., 1996*; *Ankerst et al., 1999*) was used to search for sets of touches with similar characteristics (see 'Materials and methods'). To minimize effects of adaptation (*Wang et al., 2010*) we considered only touches that occurred after long inter-touch intervals (>250 ms). We binned the remaining touch events into five groups with similar touch strength and velocity at touch.

Since the vast majority of spikes occur in a narrow time window after touch, we computed FF in a sliding window around touch (10 ms; *Figure 6A,B*). The smallest possible FF is not zero in general because spike counts are whole numbers, whereas mean spike rates are continuous. For example, for a mean spike count equal

*Figure 4. Continued*

A one-dimensional linear decoder was trained to discriminate neural activity during touch and non-touch epochs. Sets of neurons inside C2 decode touch presence with high confidence. Shading indicates 95% bounds. (**B**) Decoding of time of touch using a naïve Bayes classifier. A decoder was trained to classify neural responses occurring at different times from touch onset (the decoder assumes that the touch onset is known). Mean prediction (red line), 95% bounds (light red), true time (black dash). Inset, precision of time of touch decoding as a function of population size and time from touch. (**C**) Decoding of whisking phase using a naïve Bayes classifier. Median phase resolution from 100 decoding runs (dark lines), 95% bounds (light bands). Performance of d′ = 1 is equivalent to 0.76 of estimates falling within the resolution width. Decoding performance of whisking phase is poor even with N = 1000 neurons. Inside C2, red; outside C2, grey.

The following figure supplement is available for figure 4:

**Figure supplement 1**. Temporal decoding error.

to 0.5/touch, the minimum FF is produced with one spike in half of the trials (*DeWeese et al., 2003*). For mean spike count <1 the minimum FF corresponds to binomial spiking, with FF = 1- mean spike count (*Berry et al., 1997*). Across the population of neurons within C2 the FF following touch dropped below one, with the FF lying close to the binomial limit for most cells (*Figure 6C,D*).

Why is the FF close to 1 before and after touch (*Figure 6B*)? A brief (10 ms) sliding window was used to compute the FF, ensuring that mostly touch-related spikes were counted after touch. Since the spike rate is very low outside of the touch window, the mean spike count is also very low (~0.03), implying FF of ~1 for both the binomial and Poisson models. In contrast, around the peak of the touch response the mean spike count is high (~0.7), allowing us to detect spiking statistics that differ from the Poisson distribution.

One possible explanation for the low FF after touch is a Poisson process with a refractory period (*Berry II and Meister, 1998*). A Poisson spike train with deletion of spikes that occur during the refractory period exhibits a FF less than one. We calculated FFs for simulated spike trains with Poisson rates matched to each recorded neuron and a median refractory period of 2.3 ms (*Figure 2—figure supplement 1*). Comparing the simulations with actual spike trains revealed that L4 neuron response precision could not be explained by the refractory period alone (*Figure 6D*).

We calculated the FF with different degrees of alignment to fine-scale behavior (*Figure 6E,F*). Response variability was very large (FF = 6.57 mean ± 5.41 s.d.) when we ignored the millisecond time scale of behavior (*Figures 6E and 1*). Integrating activity in time windows as wide as average touch response periods (38 ms) gave a FF of 1.59 ± 0.49 (*Berry II and Meister, 1998*) (*Figures 6E, 2*) when randomly sampling from the trial, and FF of 1.37 ± 0.74 (*Figures 6E, 3*) when aligned to touch onset. Using a narrower touch window of 10 ms reduced the FF to 0.73 ± 0.19 (*Figures 6E, 4*). The FF was further reduced to 0.54 ± 0.27 when calculated only across similar touches (*Figures 6E, 5*). Together, these results show that excitatory neurons in L4 of barrel cortex exhibit almost noiseless responses to touch.

## Discussion

We measured the encoding of information by L4 neurons in the somatosensory cortex during active tactile sensation. Spike rates were low except for several milliseconds after touch onset (*Figures 1, 2*). During object localization, the majority (>70%) of spikes were temporally coupled to touch onset (*Figure 3*). Whisker movements organized the remaining spikes so that they aligned with particular phases of the whisk cycle. The time-scale of temporal modulation (approximately 10 ms) was much shorter than the mean inter-spike interval (approximately 1 s). Based on observations of whisker behavior it is possible to predict brief time windows when a neuron will fire a single or small number of spikes, as well as time periods when the spike probability is zero. Touch times could be reliably and precisely decoded by pooling activity from a handful of L4 neurons (*Figure 4*) (*Panzeri et al., 2014*). Spike count variance after touch (*Figure 5*), measured using the FF, was close to the binomial limit, the theoretical minimum (*Figure 6*). Based on these criteria we conclude that L4 responses encode touch with millisecond timescale precision and minimal noise.

This picture of low noise, rapidly modulated responses differs from the conclusions based on recordings from the cortex of behaving non-human primates (*Tolhurst et al., 1983*; *Shadlen and Newsome, 1998*; *Maimon and Assad, 2009*). Even in cases with time-varying stimuli and corresponding cortical responses with rapid modulation, spike counts vary greatly across trials, resulting in large FFs (*Bair and Koch, 1996*; *Buracas et al., 1998*). Four experimental factors might

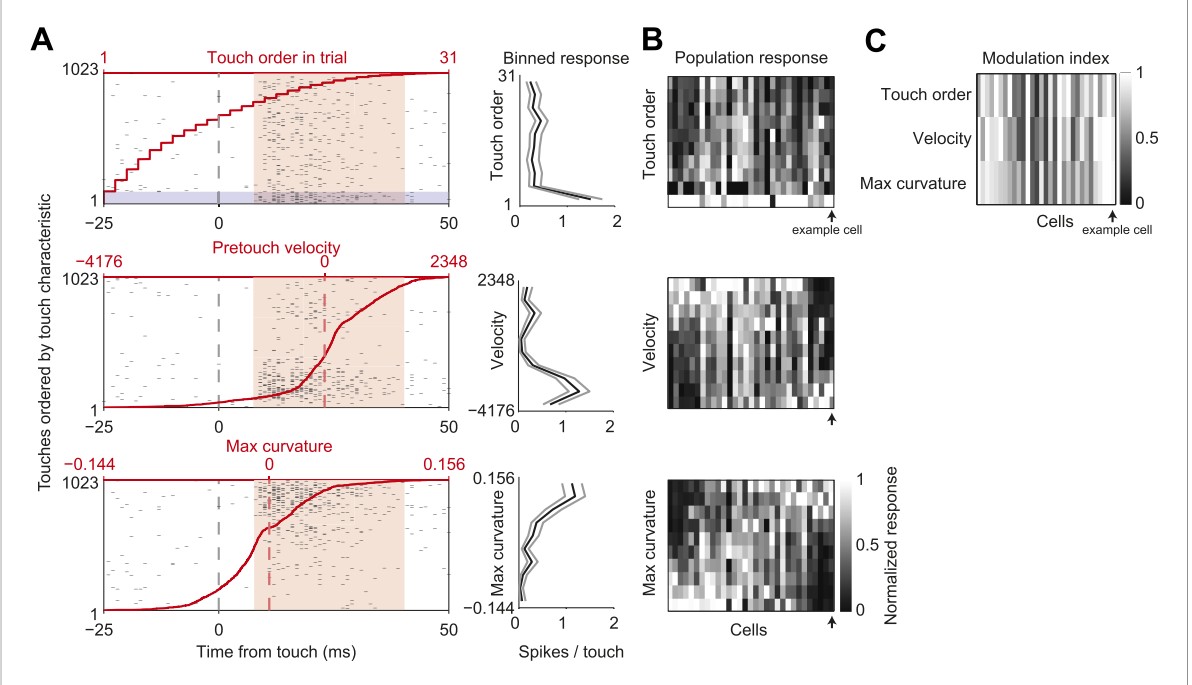

**Figure 5**. L4 spike count varies with touch properties. (**A**) Left, touch aligned spike rasters for a single cell, sorted by one of three touch properties: Order of touch in trial (top), whisker velocity at touch onset (middle), maximum whisker curvature during touch (bottom). Value of touch property corresponding to the touch (red line). Spike integration window for binned touch response (pink), first touches in trials highlighted (grey). Right, average spikes per touch for a binned range of touch property (10 bins with equal number of touches) (black line), 95% confidence interval (grey line). Same example cell as in *Figures 1, 3*. (**B**) Heatmap of the response of each L4 excitatory cell inside C2 (n = 31) to the three touch characteristics across 10 equal element bins. Responses normalized to peak for each cell. Cells are ordered by the mean tuning to maximum touch curvature. Example cell highlighted by black arrow. (**C**) Heatmap of the modulation index of the same cells and touch characteristics (max bin − min bin)/(max bin + min bin).

contribute to the large spike rate modulation and small FFs observed in our experiments: First, mice solved a discrimination task using active sensation (*Knutsen et al., 2006*; *O'Connor et al., 2010*; *Kleinfeld and Deschenes, 2011*; *O'Connor et al., 2013*). In contrast to passive presentation of stimuli (*Tolhurst et al., 1983*; *Shadlen and Newsome, 1998*; *Maimon and Assad, 2009*), in our task mice control sensory input by palpating the object with their whiskers. It is likely that mice tune their movements to achieve high signal-to-noise ratio encoding of tactile information. Second, by employing loose-seal cell-attached recordings we targeted neurons independent of activity, permitting accurate sampling of the spike trains produced by the L4 neuron population (*DeWeese et al., 2003*; *O'Connor et al., 2010*). These recording methods could be critical because standard extracellular recordings can have problems detecting neurons with low spike rate and high synchrony (*Lewicki, 1998*; *DeWeese et al., 2003*). During highly synchronous events, such as the population volley after touch, the probability is high that simultaneously recorded units in an extracellular electrode will be missed or misassigned (*Cotton et al., 2013*). Close to the binomial limit relatively few misassigned spikes can have a large impact on calculations of the FF (*DeWeese et al., 2003*). Third, compared to measurements from other cortical layers and regions, L4 spike trains in barrel cortex are more easily interpreted because extrinsic input arises mainly from the sensory periphery, rather than other cortical layers or higher cortical areas that might provide input with unobserved dynamics (*Figure 1A*).

Fourth, we track sensory input and whisker movements with temporal resolutions that are high compared to the inter-spike intervals of L4 cells. This is necessary to uncover possible influences of behavior on individual spikes. Indeed, L4 spikes appear irregular when aligned to the sample epoch (coefficient of variation >> 1, *Figure 1C,D*; *Figure 6E,F*). Only after aligning spikes to the fine-scale structure of the sensory input does the meaning of individual spikes become clear (*Berry et al., 1997*; *Baudot et al., 2013*).

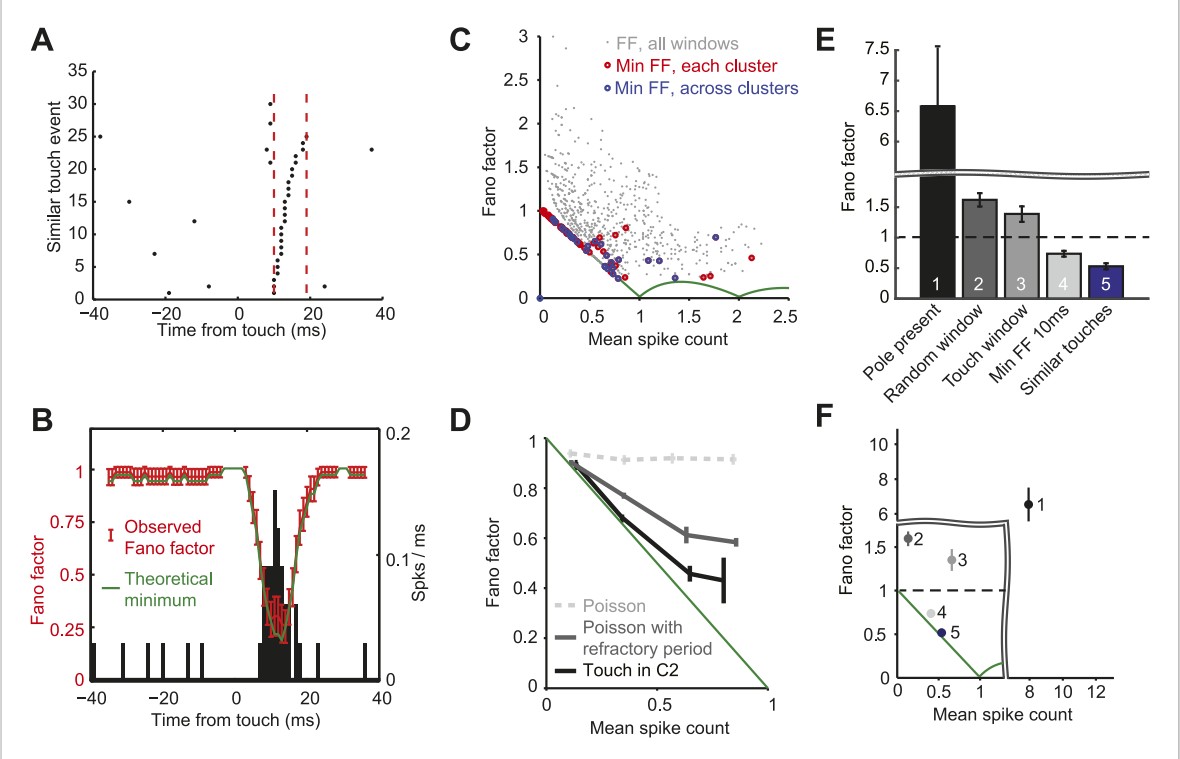

**Figure 6**. L4 responses show minimal spike count variance. (**A**) Raster plot of an example neuron aligned to touch onset. Example integration window (dashed lines) in which the neuron elicits 0 or 1 spike (black dots) per touch. (**B**) Fano factor computed over a sliding window of 10 ms (red; same neuron as **A**). Fano factor is ~1 before touch occurs because the mean spike count is very low (~0.03), which implies a minimum possible Fano factor of ~1. Error bars, bootstrap s.e.m. Theoretical minimum Fano factor (green). PSTH aligned to touch, 1 ms bins (black). (**C**) Fano factor as a function of mean spike count for all L4 neurons in C2. For each cell we calculated the Fano factor in sliding windows of 10 ms and for each of the five similar touch groups. Fano factor for each 10 ms sliding window starting from touch onset up to 20 ms post touch in 1 ms increments (grey dots). The minimum Fano factor between 0–20 ms post touch (five touch clusters per cell; red circles). The minimum Fano factor across the five groups (blue circles). Theoretical minimum Fano factor (green line). (**D**) Fano factor averaged across the population of L4 neurons (black). Fano factor expected for Poisson neurons with equivalent spike rates (dashed grey) and with a 2.3 ms refractory period (dark grey). Error bars represent s.e.m. (**E**) Comparison of Fano factors. '1', counting spikes during the sample period when the pole is within reach. '2', counting spikes in random windows of 38 ms duration. The number of epochs per trial was matched to the number of touches in each trial. '3' counting spikes in 38 ms windows after touch plus a latency of 6 ms. '4', minimum FF using a sliding window of 10 ms after touch (between 0–20 ms after touch). '5', same as '4', except touches were divided into five groups (as in panels **A**–**D**). Bars are s.e.m. (**F**) Fano factor as a function of mean spike count for the five conditions shown in panel **E**. Bars are s.e.m.

The following figure supplement is available for figure 6:

**Figure supplement 1**. Grouping of touch events using density-based clustering (OPTICS algorithm; see 'Materials and methods') for an example neuron.

In contrast to L4, only a small fraction of spikes in L5 neurons in the barrel cortex can be interpreted in terms of somatosensory behavior (*Figure 3*). This is likely because these neurons receive multiple types of input with unobserved dynamics representing hidden states (*Petreanu et al., 2009*). The irregularity of the L5 spike trains, as well as spike trains in other cell types and brain areas, may reveal themselves as deterministic fine-scale structure once the multitude of their inputs can be simultaneously monitored (*Gomez et al., 2013*).

The irregularity of cortical spike trains has often been interpreted as an irreducible feature of cortical discharges, or noise (*Renart and Machens, 2014*). This in turn has led to the view that only spike rates averaged across neuronal populations, but not precisely timed spikes, can be used to perform computations in cortical circuits (*Mazurek and Shadlen, 2002*; *London et al., 2010*). However, because a large number of inputs converge on single cortical neurons, noisy discharges are difficult to achieve in most models of neural networks (*Softky and Koch, 1993*). One exception is balanced networks, which have gained prominence in part because they produce irregular discharges

as a result of chaotic dynamics (*van Vreeswijk and Sompolinsky, 1996*; *Litwin-Kumar and Doiron, 2012*). L4 neurons have variability close to the theoretical minimum. This implies that models of cortical networks should not explicitly aim to produce intrinsically noisy activity.

Other measurements in the barrel cortex have found highly variable responses to passive whisker deflection in L4 of anesthetized rats (*Wang et al., 2010*; *Bale et al., 2013*). Several key differences between this study and ours could underlie this discrepancy. During anesthesia cortical activity is modulated by slow rhythms (e.g., up and down states), which are not observed during active behavior (*Crochet and Petersen, 2006*). These rhythms are expected to increase spike count variability. In addition, the movements underlying active sensation might recruit circuit mechanisms that reduce variability. In our experiments mice 'choose' temporally sharp touches, which are expected to drive strong and rapid feedforward inhibition within L4 (*Gabernet et al., 2005*). The inhibition shortens L4 responses after touch and thus might produce effectively binary (0 or 1 spike) responses and low variability (*DeWeese et al., 2003*). Attentional mechanisms are also known to reduce neuronal variability (*Mitchell et al., 2007*, *2009*) and may contribute to the low FFs we report here.

Neural coding of the timing of touch onset and whisking phase are key components of models of object localization (*Curtis and Kleinfeld, 2009*; *Kleinfeld and Deschenes, 2011*). We show that touch onset is reliably decodable from a small number of L4 neurons. In contrast, although many L4 cells are modulated by whisking phase, their low spike rates during whisker movements (mean <0.15 spikes/whisk cycle) hinders efficient decoding of whisking phase from population activity (*Figure 4*). The poor encoding of whisking phase is consistent with the observation that mice do not use the timing of L4 activity relative to whisking phase to measure object location, at least in the context of the simple task used in our experiments (*O'Connor et al., 2013*). How the brain uses temporally precise and low variance coding of touch in L4 neurons for tactile sensation remains to be discovered.

## Materials and methods

### Animals

All procedures were in accordance with protocols approved by the Janelia Farm Research Campus Institutional Animal Care and Use Committee. We report recordings from a total of 21 mice. 52 loose-seal cell-attached recordings were made in the following mice (7 recordings were reported in [*O'Connor et al., 2013*]): 21 recorded neurons from 7 C57BL/6 mice, 6 recorded neurons from 3 VGAT-ChR2(H134R) mice (i.e., *Slc32a1-COP4\*H134R/EYFP*) (*Zhao et al., 2011*), 17 recorded neurons from 8 PV-ires-cre mice (i.e., *Pvalb*[tm1(cre)Arbr]) (*Hippenmeyer et al., 2005*), 8 recorded neurons from 3 PV-ires-cre X Ai32 mice (i.e., *Gt(ROSA)26Sor*[tm32.1(CAG-COP4\*H134R/EYFP)Hze]) (*Madisen et al., 2012*).

### Behavior and videography

A detailed description of the behavioral apparatus, headplate installation, water-restriction schedule and training paradigm has been described (*O'Connor et al., 2010*; *Guo et al., 2014*). Mice were trained on a whisker-based go/nogo object localization task (*O'Connor et al., 2010*; *O'Connor et al., 2013*; *Guo et al., 2014*). A 0.5 mm diameter pole (class ZZ gage pin, Vermont Gage) was presented in one of two locations, 4–8mm apart on the anteroposterior axis. Mice licked the spout of an optical or electrical lickport to receive water reward if the pole was located in the posterior position, and withheld licking in the anterior position. No airpuff punishment or active removal of residual water from the lickport was used. Mice were trimmed to a single C2 whisker to perform the discrimination.

For each behavioral trial whisker video was recorded for 4–5 s, spanning the period prior to pole movement and following the response window (*Clack et al., 2012*; *Pammer et al., 2013*). Video frames were acquired in Streampix 3 software (Norpix, Canada) at 1000fps with 90–200 µs exposure times (Edmunds Optics #58-257). Videography was with a 0.36× telecentric lens and a Basler 504k camera under 940 nm LED illumination (Roithner Laser). Whisker trajectories and shapes were automatically quantified using the Janelia Whisker Tracker (https://openwiki.janelia.org/wiki/display/MyersLab/Whisker+Tracking; [*Clack et al., 2012*]). Contact periods with the pole were automatically determined by pole proximity and whisker curvature using custom Matlab routines (https://github.com/hireslab/HLab_Whiskers) and manually curated to ensure 1 millisecond accuracy.

## Whisker analysis

The behavioral time-series were separated into touch, whisking, and non-whisking epochs. Touch epochs were periods where the whisker was in contact with the pole. Whisker curvature was measured, with $K = 1$/radius of the osculating circle tangent to the point of curvature measurement (*Pammer et al., 2013*). For other epochs, time-series of the azimuthal angle (theta) were bandpassed between 6–60 Hz (Butterworth fourth order) followed by decomposition by the Hilbert transform (*Hill et al., 2011*). Whisking epochs corresponded to periods where the zero-crossing phase of the Hilbert transform had amplitudes >2.5°, during which individual whisking cycles spanning –π to π were extracted. Whisking cycles during licking or within 70 ms after touch were excluded. Whisking epochs used for phase analysis had monotonic phase for a complete whisking cycle. Non-whisking periods were defined as contiguous periods of at least 100 ms with no touch or licking and with whisking amplitude <1.25°. Reaction time is the time between the first touch onset and the first lick calculated for every trial where licks occur (*Figure 1—figure supplement 1*).

## Electrophysiology

On the day of the first recording, a small craniotomy (~200 μm diameter) was made over the C2 barrel column determined by transcranial intrinsic signal imaging (*O'Connor et al., 2010*). The dura was left intact. Recordings targeting cortical L4 were obtained with patch pipettes pulled from borosilicate tubing (Sutter instrument, CA) and an Axopatch 700B amplifier (Molecular Devices). Loose-seal juxtacellular pipettes were filled with ACSF or cortex buffer (in mM): 125 NaCl, 5 KCl, 10 dextrose, 10 HEPES, 2 CaCl$_2$, 2 MgSO4, pH 7.4, osmolality ~272 mmol/kg. The manipulator depth was zeroed upon pipette tip contact with the dura (*O'Connor et al., 2010*). After contact, the craniotomy was covered by cortex buffer or 2% agar in cortex buffer. Aided by positive pressure (1 PSI), the pipette was advanced through the dura. When searching for cells, the pipette pressure was reduced to 0.1–0.3 PSI. Two pipette shapes were used, with a thicker shank (3.5–5 MΩ) (*O'Connor et al., 2010*) or a thinner shank (6–9 MΩ). Cells were recorded blindly (*DeWeese et al., 2003*). Recordings using thick shank pipettes caused cortical dimpling of ~100 μm (*O'Connor et al., 2010*). Cells recorded with thick shanks and raw manipulator depths of 505–665 μm (405–565 μm corrected; 11 cells) were considered L4 and depths of 727–944 μm (627–844 μm corrected; 11 cells) were considered L5. Cells recorded with thin shank pipettes with raw manipulator depths of 444–560 μm (uncorrected; 30 cells) were considered L4 (*Figure 3—figure supplements 1, 2*). This depth range was consistently in L4 based on juxtacellular cell fills (unpublished observations). Data acquisition was controlled by *Ephus* (*Suter et al., 2010*). The sampling rate was 10 kHz.

Following the final recording, a DiI coated pipette was inserted into the craniotomy. Recordings in L4 with the midpoint of the DiI track <160 μm from the center of C2 were considered in C2 (31 recordings) and those >200 μm but still in the barrel field (10 recordings) were considered outside of C2 (*Figure 1—figure supplement 2*). L5 recordings ranged 98–324 μm from the center of C2 (near C2). Recordings with unstable spike rate across the behavioral session were excluded. Recordings were repeated for 1–5 days per animal.

## Histology

Mice were deeply anesthetized with 5% isoflurane then perfused with 0.1 M sodium phosphate buffer followed by 4% paraformaldehyde (PFA, in 0.1 M phosphate buffer, pH 7.4). The brain was immersed in fixative for at least 24 hr before sectioning. The fixed cortex was flattened and fixed, and 100 μm slices were cut tangentially. Cytochrome oxidase (CO) staining was performed to reveal the barrel field (*Land and Simons, 1985*) and fluorescence imaging to determine the DiI pipette track relative to the field (*Figure 1—figure supplement 2*).

## Spike analysis

For touch analyses, peri-contact time histograms of spikes (PCTHs), aligned to either first touch or later touch onset, were constructed (*Figure 3—figure supplements 1, 2*). The peak touch rate (*Figure 2*) was the highest mean spikes per bin (1 ms) of the cell's PCTHs, typically corresponding to the first touch. Spike onset latency is the time between the touch onset and where the rise of the PCTH exceeds the pretouch mean (−50 to 0 ms pre-onset) by two standard deviations (*Figure 2—figure supplement 1*). Spikes evoked per touch is the mean spike count within the touch-onset coupled spike window,

defined below (*Figure 5*). For phase analysis, spikes were aligned to the whisking phase by linearly interpolating spike time to phase. For each neuron we built a histogram with 12 bins and fitted to a cosine function: $A[1 + \cos(\theta - \varphi_{pref})] + B$ with A, B > 0 (*Figure 3—figure supplements 1, 2*). From the fit we extracted the preferred phase ($\varphi_{pref}$) and the modulation depth $A/(A + B)$ (*Figure 2—figure supplement 1*).

## Spikes accounted by active sensation

We estimated the proportion of spikes attributed to touch onset and whisking phase during active exploration (*Figure 3*). Active exploration epochs are whisking epochs, including touch epochs (excluding prolonged touches, > 100 ms). For touch onset, each spike was indexed by the time elapsed since the closest touch onset (taking into account the latency of each cell). Each neuron was characterized by a curve, $y_{ton}(t_i)$, representing the total number of spikes that occurred within a given time window ($t_i$ with $i = 0, 1 \ldots etc$). In cells where most spikes occurred shortly after touch onset these curves rise steeply and then plateau. For each neuron we determined the touch response cutoff, the first time point $t_i = t_{on}$ in which the rate of increase $\Delta y_{ton}(t_i) = y_{ton}(t_i) - y_{ton}(t_{i-1})$ is below chance level of the overall firing rate (p < 0.05; bootstrap method). The chance level was estimated by shuffling the spike times (1000 repetitions) to create surrogate curves $y_{ton}^k(t_i)$. The derivative was calculated from these curves smoothed with a polynomial fit (degree 11). For each $t_i$ we estimated the 95% percentile of the slope of the surrogate curves. For whisking phase, each spike that was not attributed to touch onset was indexed by the time elapsed between the spike and the closest whisk cycle preferred phase (taking into account phase circularity and including whisking epochs with non-monotonic phase). As with touch, we built $y_{whisk}(t_i)$ representing the number of spikes that occurred around the preferred whisking phase. We determined the whisking spike cutoff, the first time point $t_i = t_w$ in which the rate of increase of explained spikes by whisking was below chance levels.

## Population decoding

To decode touch based on spikes for each neuron '$i$' (*Figure 4*) we extracted the spiking response $r_{i,j}(t)$ of all touch events '$j$' from $t = 30$ ms before touch up to $t = 50$ ms after touch (with $j = 1 \ldots N_{touch}(i)$; excluding events with a second touch in the 50 ms post-touch window). We randomly selected a set of neurons (up to N = 200) and for each neuron we randomly sampled (with replacement) 1000 of its touch aligned responses $r_{i,j}(t)$. Each spike train was causally integrated with a defined window (i.e., w = 10 ms). At each time point $t_j$ we built a different decoder that was trained to discriminate between the pooled population response at $t_j$ and the population response at epochs without touch (i.e., a one-dimensional decoder). We used half of the data to find the optimal threshold and the other half of the data to predict the performance of the decoder. Other decoders, such as naïve Bayes classifier and Fisher linear discriminant, produced similar results (not shown). The decoders assume implicit knowledge of when the touch occurred, since at each time point a different decoder was employed.

To decode the time elapsed since touch onset we used the same sampling procedure as described above and integrated the response with a sliding window with duration of 10 ms. We trained a multinomial naïve Bayes classifier (*Duda et al., 2001*) to report for every time point what was the most likely time elapsed from touch onset, with a root mean square time error derived from boostrapping (100 runs) (*Figure 4—figure supplement 1*). To decode the whisking phase we assumed that the preferred phase was uniformly distributed. For each neuron we aligned the spikes to the whisking phase of each whisking cycle (minimum peak whisking amplitude >2.5°) and binned the response in 120 bins between $\pi$ and $-\pi$. To simulate the population response we randomly picked a set of neurons (with replacement) and circularly shifted their response to obtain a uniform distribution of preferred phases. We trained a naïve Bayes classifier to discriminate the population response to two different whisking phases. We tested the performance of the decoders in function of the difference of whisking phase and determined the phase difference that achieved 76% correct discrimination in function of the number of neurons.

Under the assumption of independent neural responses, our simulations show several-fold more information about touch in C2 than non-C2 and poor performance in decoding whisking phase (*Figure 4*). Other factors, such as correlated activity among neurons, can impact the accuracy of the population code (*Dayan and Abbott, 2001*; *Moreno-Bote et al., 2014*) and the exact information carried by touch and whisking phase, but this is unlikely to alter the qualitative picture presented here.

## Analysis of spike count variance

To sort out external and intrinsic contributions to spike count variability, we took several steps to reduce external variability due to differences in sensorimotor variables. Touch responses in L4 neurons are modulated by adaptation, pretouch velocity and curvature of the whisker (which is proportional to touch force) (*Figure 5*). To reduce adaptation effects we selected touch epochs in which the inter contact interval (ICI) was longer than 250 ms. We divided the remaining touch events into five groups (*N*, number of points per group; minimum, 20), clustered by touch characteristics. We z-scored pretouch velocity and the maximum curvature change shortly after touch (0–20 ms). For each neuron we clustered the touch events using a density-based clustering method (OPTICS; [*Cunningham and Yu, 2014*]). The output of OPTICS gives an ordered list $l_i$ of points sorted by similarity. We searched for the set of consecutive sorted points $l_i, l_{i+1}, \ldots, l_{i+N-1}$ that mimimized the sum over all pairwise distances. After obtaining the set of touch events for the first bin, we removed those points and proceeded in the same manner to obtain the second data bin. We repeated the procedure until obtaining five data bins (*Figure 6—figure supplement 1*). We calculated the FF by counting spikes in sliding windows of 10 ms for each cell and each of the five bins (*Figure 6*). Confidence intervals for the FFs were obtained by resampling 1000 times. We also computed FFs using a Poisson process with absolute refractory period, with average touch response matched to the data (*Berry II and Meister, 1998*) (*Figure 2—figure supplement 1*).

## Acknowledgements

We thank Mike DeWeese, Shaul Druckmann, David Golomb, Judith Hirsch, Máté Lengyel, Nuo Li, Simon Peron, Alex Pouget, Sandro Romani, Nick Sofroniew, and Fritz Sommer for comments on the manuscript.

## Additional information

### Funding

| Funder | Author |
| --- | --- |
| Howard Hughes Medical Institute (HHMI) | Samuel Andrew Hires, Diego A Gutnisky, Jianing Yu, Daniel H O'Connor, Karel Svoboda |

The funder had no role in study design, data collection and interpretation, or the decision to submit the work for publication.

### Author contributions

SAH, Conception and design, Acquisition of data, Development of Whisker analysis code, Drafting the article; DAG, Conception and design, Development of Whisker analysis code, Drafting the article; JY, Acquisition of data, Analysis and interpretation of data, Final approval of article; DHO, Conception and design, Development of Whisker analysis code, Final approval of article; KS, Conception and design, Development Whisker analysis code, Drafting the article

### Ethics

Animal experimentation: All procedures were in accordance with protocols approved by the Janelia Farm Research Campus Institutional Animal Care and Use Committee (#11-71).

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
