## [Decision Letter]

Thank you for sending your work entitled “Low-noise encoding of active touch by layer 4 in the somatosensory cortex” for consideration at *eLife*. Your article has been favorably evaluated by a Senior editor and three reviewers, one of whom is a member of our Board of Reviewing Editors.

The Reviewing editor and the other reviewers discussed their comments extensively before we reached this decision, and the Reviewing editor has assembled the following comments to help you prepare a revised submission.

1) This revised submission does not require new experiments, but it does require new analyses and ideally new data from other cortical layers, which the authors may already have in their database.

This is potentially a solid study that sheds some light on the factors that determine what makes cortical neurons fire. The paper reports on the variability of spike trains in layer 4 (L4) neurons in mouse somatosensory cortex during a touch task (pole localization) involving active whisking. It makes a convincing argument that variability is very low, once spikes are aligned properly based on time of touch and on whisk phase. Specifically, touch (more than whisking phase) seems to be the primary feature encoded by the timing of the spikes. Indeed, most spikes can be explained by whisker curvature and related kinematics, together with (to a smaller degree) whisking itself. Around the time of the touch the noise in these neurons is near the theoretical minimum. Thus what might appear as trial-to-trial variability or noise is not. Spiking appears to be noisy only if the observer does not or cannot take into account the true determinants of spiking.

2) This paper, however, doesn't really change our view of sensory processing in L4 of barrel cortex. It shows that L4 responses can be predicted rather closely if one knows a few fundamentals about the sensory and motor time courses. This is hardly surprising: it would have been surprising if one could have predicted spike trains without knowledge of those fundamentals. Still, it is interesting to see that most spikes can be accounted for, once one has those simple pieces of information.

3) This said, the study also suffers from major limitations. Principally, the main finding is not novel – it is made to appear novel only by adopting strawman hypotheses, ignoring much of the relevant and recent literature.

4) Indeed, it is widely agreed that individual cortical neurons are not noisy: if cortical responses are variable, the variability is due to integration of inputs, which is mostly seen in central neurons.

5) Moreover, as the paper points out, layer 4 in this system is close to the sensory periphery, with little input from the rest of cortex – and we already know that in the sensory periphery there is very little noise. These and other criticisms are laid out below.

6) Premise of the paper. The Introduction begins with “Cortical spike trains are often considered to be noisy”, and casts much of the paper in the context of this view. But after decades of work, this idea of cortical neurons being “noisy” has become a strawman. It is out of step with crucial pieces of the literature. We already know that cortical neurons are not intrinsically noisy; see e.g. Mainen and Sejnowski, “Reliability of spike timing in neocortical neurons”. Science 1995, and Deweese and Zador, “Shared and private variability in the auditory cortex”. J Neurophysiol 2000. Also, the apparent Poisson nature of variability can be explained by a deterministic spike threshold (Carandini, Plos Biol 2000). Similarly, the statement that “In cortex, irregular spike trains suggest that only rate codes can be used to perform reliable computations” is misleading, and based on a reference that is over 10 years old.

7) Vision literature. The paper uses the literature from visual cortex to make the point that cortical responses are noisy. But even in that literature there is much that contradicts this point. In a paper titled “Low response variability in simultaneously recorded retinal, thalamic, and cortical neurons” (Neuron, 2000), Kara, Reinagel, and Reid have already shown that responses in layer 4 of visual cortex can be remarkably repeatable. And these results are not limited to anesthetized preparations: similar results have been obtained in awake monkeys, where (as one could have imagined, and similarly to what is done here for whiskers) it is essential to know eye position on a moment by moment basis: Gur, Beylin, and Snodderly, “Response variability of neurons in primary visual cortex (V1) of alert monkeys” (J Neurosci, 1997).

8) Somatosensory literature. If we now turn to the recent literature on somatosensory cortex, we see that it makes precisely the opposite case relative to neurons being “noisy”. Indeed, the paper might just as well have begun with: “Cortical spike trains are often considered to be made up of precisely timed spikes, whereby the timing of each spike carries some meaning about the stimulus.” See for example Arabzadeh, Zorzin and Diamond “Neuronal encoding of texture in the whisker sensory pathway” (PLoS Biol 2005) and “Deciphering the spike train of a sensory neuron: counts and temporal patterns in the rat whisker pathway” (J Neurosci 2006). Additional contributions that should be considered in rethinking how to cast these results include those from G. Stanley (admittedly anesthetized, as noted in this study), E. Arabzadeh, G. Foffani, R. Petersen, and S. Panzeri. The latter include “Information carried by population spike times in the whisker sensory cortex can be decoded without knowledge of stimulus time” (Frontiers, 2010) – which is related to the authors' conclusion that “Pooling activity from only fifteen L4 neurons […] in C2 was sufficient to detect 95 % of touches” – and “Complementary Contributions of Spike Timing and Spike Rate to Perceptual Decisions in Rat S1 and S2 Cortex” (Current Biology, 2015). The latter is admittedly too recent to have shaped the present paper, but it is now in the literature.

9) Poisson spike generators. Given the literature, the Poisson model with a refractory period has become a strawman. Therefore, the simulation of Poisson firing with refractory period (in the sixth paragraph of the subsection “Low spike count variability”) is unsatisfying. We already know that the Poisson model is not a reasonable model, even with refractory periods. We know this in retina, in LGN, in V1, and in A1. It would be more interesting to test a more plausible model. For instance, would an integrate-and-fire model with Gaussian noise in Vm give the observed statistics?

10) Fano Factor. Related to the last point: is it sensible to calculate Fano Factor (FF) for such a radically non-stationary process? Fano Factor is at best awkward for non-stationary processes and the main message of the paper is that spiking is non-stationary, e.g. extremely variable over large time scales and extremely precise when whiskers kinematics dictate so. For a Poisson process, FF = 1, independent of the window size. Since the observed FF for full stimulus window was large, evidently a Poisson process is not underlying spikes. At best it is a Poisson process driven by a variable mean rate. Indeed, Poisson statistics are meant to model stationary processes, that is, processes where the probability of an event per unit time is constant. Obviously firing probability increases in the tens of ms after touch, compared to the rest of the trial.

11) Further, having demonstrated that the neurons of interest in barrel cortex do not resemble a Poisson process, the modeled neurons do: “We used a simplified model based on independent Poisson neurons with rapidly modulated spike rates”. It is puzzling to model Poisson neurons once one knows that the model is not correct.

12) Factors accounting for spike trains. In the present analysis (in the second paragraph of the subsection “Low spike count variability”), 3 factors are used to account, one at a time, for spike trains: order of touch, whisker velocity, and whisker curvature. Why use them one at a time? Why not use all 3 at once and see how much better one does? Or use only the two main ones? Are there any spikes unaccounted for once one does this? This could be done with a GLM (perhaps enlisting the help of colleague Jeremy Freeman) or with similar approaches. Indeed, since the main message of the paper is that spike trains can be explained by whisker curvature and related kinematics, it would make sense to place less emphasis on the Poisson analysis and more emphasis on showing whisker state as a determining factor.

13) Other layers. Rather than focusing solely on L4, it would have been much more interesting to compare L4 with other cortical layers, and investigate the sources of variability there. Indeed, a major missing component is that neurons outside L4 are not recorded or reported. This is set up in the Introduction, but not really addressed. It's great to show L4 does one thing, but more interesting to show that it does something different than other layers. Given other publications from this group, it seems likely that the authors have other data that addresses this. Analyzing these data would make the paper much stronger.

---

## [Author Response]

*1) This revised submission does not require new experiments, but it does require new analyses and ideally new data from other cortical layers, which the authors may already have in their database*.

The reviewers are presumably referring to the database from O’Connor et al. 2010. Unfortunately, in this previous study and all other previous studies from our lab and other labs, the behavioral tracking (videography) was not sufficiently accurate to extract touches with millisecond precision and low error rates.

However, we agree that the paper would be stronger by comparing our results with other neuron types that are known to receive long-range cortico-cortical inputs. One of the main points of our work is that virtually all L4 spikes can be explained based on behavior. The layer 4 circuit in one S1 barrel receives external inputs mainly from ∼200 neurons of the corresponding barreloid in VPM. Other neuron types, such as layer 5 pyramidal neurons, receive in addition inputs from diverse sources that we do not observe (motor cortex, S2 etc); this can cause apparently variable neural responses.

To enable a comparison between microcircuit types, we have extended our analysis to a new set of L5 neurons. In contrast to L4, only a small percentage of spikes in L5 can be explained by whisking and touch. The new L5 data has been included in Figure 2 and Figure 3.

We have extended our L4 data set as well, adding 14 new recordings in L4 C2 bringing the total from 17 neurons to 31 neurons.

*2) This paper, however, doesn't really change our view of sensory processing in L4 of barrel cortex. It shows that L4 responses can be predicted rather closely if one knows a few fundamentals about the sensory and motor time courses. This is hardly surprising: it would have been surprising if one could have predicted spike trains without knowledge of those fundamentals. Still, it is interesting to see that most spikes can be accounted for, once one has those simple pieces of information*.

We respectfully disagree with this point. The experimental literature overwhelmingly reports highly variable neural responses (e.g. references in our manuscript; Carandini, 2004; [13]; Goris et al. 2014). This has influenced theoretical modeling in such a way that variable (Poisson-like) spiking statistics are a measure of success for networks models of the brain (e.g. van Vreeswijk & Sompolinsky, 1996; Litwin-Kumar & Doiron, 2012).

Although it may not be ‘surprising’ to the reviewers that L4 spike rates and spiking statistics can be ‘accounted for’ by behavioral variables, the quantitative extent in behaving animals is new (and surprising to us).

In fact, we see little published prior art on these key points made in our paper:

A) Our measurements are in defined neurons in an active sensory perception task. Most of what we know about L4 responses in mice and rats comes from experiments in anesthetized experiments with passive stimulation. Even responses in the ganglion cells are very different with piezo stimulation compared to actively sensing rodents (Rasmus Petersen, Thalamus meeting Janelia 2015). Furthermore, anesthesia puts the brain in a non-natural state. Studies of neural coding have to be ultimately performed in a well-controlled behavioral task (Renart & Machens, 2014).

B) The existing literature clearly does not predict our results, even for simple metrics such as spike rate. For example, in the only other paper reporting recordings in L4 in awake animals (Curtis & Kleinfeld, 2009) baseline spike rates are high and modulation is relatively low – implying that a relatively small fraction of spikes would be explained by behavior. The difference might be due to recording techniques (cell-attached vs. wire electrodes), behavioral state, or tracking of the behavior.

C) Our quantitative estimate of the fraction of spikes that are behavior-related is a new analysis and observation. Our analysis goes well beyond qualitative assessments (i.e. classification as cells responding to touch or not). Importantly, our analysis could not have been done without millisecond timescale tracking of behavior.

D) The observed Fano factors are much smaller than any reported before, even though our measurements were done in active behavior. Careful tracking of behavior and best-of-class single unit isolation are critical for this analysis. There is a great interest in understanding and explaining variability in neural responses. For instance Goris et al. (2014) found that a stimulus-independent parameter can account for the super-Poissonian variability observed in multiple visual areas. Ecker at al. (2014) found that under anesthesia correlated variability is modulated by slowly fluctuating input that also significantly reduces the correlation among neurons. Our work highlights that the neural code can be close to noiseless ‘under battleground conditions’.

We have rewritten parts of the paper to make these points clearer (Introduction).

*3) This said, the study also suffers from major limitations. Principally, the main finding is not novel – it is made to appear novel only by adopting strawman hypotheses, ignoring much of the relevant and recent literature*.

We respectfully disagree with the statement that our study is not novel. To the best of our knowledge, the major take-home messages in our paper are new (also see response to Point 2).

A) The vast majority of spikes, in many cases >90%, can be accounted for in terms of tactile behavior.

B) The intrinsic variability of responses was substantially lower than expected for Poisson spike trains and often reached the theoretical minimum.

We are not aware of any studies that showed (A) and (B) in behaving animals in any cortical area.

We understand that ‘strawman’ refers to our Fano factor (FF) calculation, where we compare FF in the vicinity of touch (low) to the FF integrated over the full sampling period. This comparison could be easily removed without changing any of our conclusions. However, we feel that this comparison serves as an important didactic point.

The ‘strawman’ shows that FF can be affected dramatically if behavior is not measured with precision. The same would hold if neural dynamics is influenced by the hidden dynamics of unobserved brain areas (as is the case in most experiments where this sort of analysis has been attempted).

We have reworded the offending sections to dismantle the strawman (Results):

“Neuronal variability can arise from external factors […] at least in part due to trial-to-trial variability in active touch.”

We reply below to the point on recent literature on cortical noise.

*4) Indeed, it is widely agreed that individual cortical neurons are not noisy: if cortical responses are variable, the variability is due to integration of inputs, which is mostly seen in central neurons*.

Please note that we do not write nor imply nor state that ‘individual cortical neurons’ are noisy. In fact, in our Introduction we write:

“…noisy discharges are difficult to reconcile with the observed high reliability of cortical neurons (Mainen and Sejnowski, 1995) and groups of synapses (Stevens and Zador, 1998)”.

It is well established that intrinsically noisy networks can be constructed with reliable neurons.

We state that trial-to-trial variability in neuronal responses (e.g. spike count) has been observed in visual cortex, motor cortex, somatosensory cortex, both in behaving and anesthetized preparations (e.g. [13]). Contrary to the reviewers’ assertions, the causes of this variability are unknown and an intense area of study (for reviews see Renart & Machens, 2014). Our point is that if extrinsic factors are taken into account cortical neurons are remarkably precise (as precise as they can be!). One important corollary is that we do not need network models that are intrinsically noisy.

*5) Moreover, as the paper points out, layer 4 in this system is close to the sensory periphery, with little input from the rest of cortex – and we already know that in the sensory periphery there is very little noise. These and other criticisms are laid out below*.

In terms of number of synapses L4 is as far from the periphery as L5 and L6; all of these layers receive direct input from VPM. The key point is that L4 receives long-range input almost exclusively from VPM (and recurrent input from other L4 neurons); it’s therefore a relatively simple cortical circuit and by monitoring sensory information we monitor a lot of the input to L4. In contrast, L5 and L6 receive inputs from other long-range sources.

We cite papers in which low response variability was recorded (below; however, none during active behavior):

“However, some studies in the sensory cortex suggest that the apparent irregularity in spike trains does not reflect noise but the hidden states of a temporally rich signal (DeWeese, Wehr and Zador 2003; Gur, Beylin and Snodderly 1997; VanRullen, Guyonneau and Thorpe 2005; [2]; [38]).”

Many recent high profile papers report high FF in multiple brain areas:

[13]. FF: 1-2 in V1, V4, MT, LIP, PRR, PMd, OFC.

Carandini (2004). FF ∼= 1.

Goris et al. (2014). From LGN, V1, V2, MT. FF typically in 1-10 range.

[53]. V4, FF ∼=1.4.

Cohen and Maunsell (2009). V4, FF ∼=1.

[71]. L4 in S1 FF∼=0.8.

[1]. S1. FF = 0.8-1.2.

Bale & Petersen (2013). FF in S1 = 1.3, but close to minimum FF in ganglion (∼0.3), and still low in VPM (0.5). These results contradict the reviewers comment that as L4 is close to the periphery it is expected to exhibit also low noise.

Theoretical papers that are modeling high FF:

Litwin-Kumar & Doiron (2012). They show that slightly non-random networks cause cortical variability and stimulus-dependent quenching of neural variability. The idea that cortical neurons have FF>1 motivates this paper.

van Vreeswijk & Sompolinsky (1996). Influential paper in which a balanced state is proposed that can explain large FF of cortical neurons. This type of model has been the standard when trying to do simulations to generate poisson-like firing statistics.

Beck, et al., Pouget (2012). They suggest that Poisson noise actually reflects suboptimal inference.

By simplifying the problem (choosing a well-known, relatively simple circuit; controlled behavior; gold-standard recording techniques and appropriate analyses) we discovered very low variability. Our paper thus calls into question that cortical circuits inherently generate noisy discharges.

Recent review papers call for reassessment of the idea of cortical noise (Renairt & Machens 2014; Masquelier, 2013).

Masquelier: “High trial-to-trial variability in response to repeated presentation of a same stimulus has been reported in every modality. It is often quantified in terms of reliability and precision (Box 1), and both are usually poor in vivo (e.g. Fano factors ∼1 and precision ∼tens of ms or above). The origin of this variability, and its implication for information processing, has been much debated (Stein et al., 2005; Ermentrout et al., 2008; Faisal et al., 2008; Tiesinga et al., 2008; Rolls and Deco, 2010), yet a consensus has not emerged. Here we argue that most of the observed variability could come from uncontrolled variables, or the use of inappropriate reference times, rather than from intrinsic sources of noise (“intrinsic” meaning that they cannot be eliminated).”

We believe that our study sheds light onto these issues. The citations above clearly show that many researchers in the field feel that this is important and unsettled business.

*6) Premise of the paper. The Introduction begins with* “*Cortical spike trains are often considered to be noisy*”*, and casts much of the paper in the context of this view. But after decades of work, this idea of cortical neurons being* “*noisy*” *has become a strawman. It is out of step with crucial pieces of the literature. We already know that cortical neurons are not intrinsically noisy; see e.g. Mainen and Sejnowski,* “*Reliability of spike timing in neocortical neurons*”*. Science 1995, and Deweese and Zador,* “*Shared and private variability in the auditory cortex*”*. J Neurophysiol 2000. Also, the apparent Poisson nature of variability can be explained by a deterministic spike threshold (Carandini, Plos Biol 2000). Similarly, the statement that* “*In cortex, irregular spike trains suggest that only rate codes can be used to perform reliable computations*“ *is misleading, and based on a reference that is over 10 years old*.

Please see the responses above.

We further disagree that we are misleading with our statement that it has been argued that irregular spike trains suggest that only rate codes can be used to perform reliable computations. London et al. (2010) and others make exactly that argument. The title of their paper is: “Sensitivity to perturbations in vivo implies high noise and suggests rate coding in cortex”. And they take variability in cortical discharges for granted: “It is well known that neural activity exhibits variability, in the sense that identical sensory stimuli produce different responses”.

Mainen & Sejnowski (1995; cited) is slice work and we have no disagreements with them. Neural networks can generate noisy discharges with deterministic neurons (there are literally hundreds of papers on this). In the last 10 years nothing has happened as far as we can tell that has challenged the status quo.

We feel that we present the status quo fairly in our revised Introduction.

*7) Vision literature. The paper uses the literature from visual cortex to make the point that cortical responses are noisy. But even in that literature there is much that contradicts this point. In a paper titled* “*Low response variability in simultaneously recorded retinal, thalamic, and cortical neurons*” *(Neuron, 2000), Kara, Reinagel, and Reid have already shown that responses in layer 4 of visual cortex can be remarkably repeatable. And these results are not limited to anesthetized preparations: similar results have been obtained in awake monkeys, where (as one could have imagined, and similarly to what is done here for whiskers) it is essential to know eye position on a moment by moment basis: Gur, Beylin, and Snodderly,* “*Response variability of neurons in primary visual cortex (V1) of alert monkeys*” *(J Neurosci, 1997)*.

We cite Gur et al. (1997). They show that responses in V1 are more reliable when fixational eye movements are taken into account. This paper is indeed important prior art and is discussed as such. However, this study does not show that responses are close to the theoretical minimum (which would be FF ∼ 0 for their high spike counts). Kara et al. (2000; also cited) show that responses can be sub-Poissonian in anesthetized cats.

*8) Somatosensory literature. If we now turn to the recent literature on somatosensory cortex, we see that it makes precisely the opposite case relative to neurons being* “*noisy*”*. Indeed, the paper might just as well have begun with* “*Cortical spike trains are often considered to be made up of precisely timed spikes, whereby the timing of each spike carries some meaning about the stimulus.*” *See for example Arabzadeh, Zorzin and Diamond* “*Neuronal encoding of texture in the whisker sensory pathway*” *(PLoS Biol 2005) and* “*Deciphering the spike train of a sensory neuron: counts and temporal patterns in the rat whisker pathway*” *(J Neurosci 2006). Additional contributions that should be considered in rethinking how to cast these results include those from G. Stanley (admittedly anesthetized, as noted in this study), E. Arabzadeh, G. Foffani, R. Petersen, and S. Panzeri. The latter include* “*Information carried by population spike times in the whisker sensory cortex can be decoded without knowledge of stimulus time*” *(Frontiers, 2010) – which is related to the authors' conclusion that* “*Pooling activity from only fifteen L4 neurons […] in C2 was sufficient to detect 95 % of touches*“ *– and* ”*Complementary Contributions of Spike Timing and Spike Rate to Perceptual Decisions in Rat S1 and S2 Cortex*“ *(Current Biology, 2015). The latter is admittedly too recent to have shaped the present paper, but it is now in the literature*.

It’s been known since the classic studies of Dan Simons (cited) that after passive whisker deflection L4 neurons spike with short latencies and little timing jitter. We cite this and other papers in our Introduction.

We show similar temporal precision in L4 neurons during active sensation. Furthermore we show that spike rates are extremely low except in a narrow time window after touch. This picture is very different compared to the only other L4 recordings during active behavior that we are aware of (Curtis and Kleinfeld, 2009); this study shows high baseline spike rates and thus relatively small fraction of spikes explained.

Importantly, rapidly changing spike rates do not imply low variability! [4] have already shown temporally precise responses with large FF. Where it’s been looked at variability has been relatively high in anesthetized barrel cortex (FF ∼ 1; e.g. [1]).

We thank the reviewers for pointing out the papers from Diamond’s lab. We now discuss one of them in our revised manuscript ([60]; in the subsection “Decoding touch”).

We note in passing that Diamond too acknowledges the need to identify the sources of cortical variability (Arabzadeh et al., 2005):

“In a number of sensory modalities, first-order neuron responses can be remarkably reliable when a stimulus is presented repeatedly, whereas cortical responses vary across trials. It is of interest to elucidate the mechanisms that permit reliable first-order neuron responses and, by the same token, to identify the sources of trial-to-trial variability among cortical neurons.”

“From these observations we conclude that, under our experimental conditions, the trial-to-trial response variability of first-order neurons is caused exclusively by stimulus jitter, whereas that of cortical neurons results mainly from variations across time in sensory integration, and must emerge at some integration site between the trigeminal ganglion and cortex. A question of current interest is whether the variability in cortical responses results from noise and imprecision in neuronal integration, or else reflects functionally significant modulations in responsiveness.”

*9) Poisson spike generators. Given the literature, the Poisson model with a refractory period has become a strawman. Therefore, the simulation of Poisson firing with refractory period (in the sixth paragraph of the subsection “Low spike count variability”) is unsatisfying. We already know that the Poisson model is not a reasonable model, even with refractory periods. We know this in retina, in LGN, in V1, and in A1. It would be more interesting to test a more plausible model. For instance, would an integrate-and-fire model with Gaussian noise in Vm give the observed statistics*?

We respectfully disagree with this characterization of Poisson models. Spike trains are almost always modeled as Poisson trains.

Recent papers (e.g. Goris, Movshon, & Simoncelli, 2014) insist on the Poisson model with some variations. All GLM models we are aware of have an output that is a Poisson generator (refractoriness is obtained with a post-spike filter). Given that we observe variability below Poisson, the next simplest model would be to add a refractive period. This is a statistical, not a mechanistic model (as the reviewers propose). Our analysis says that the low variability we report is not due to very high spike rate changes followed by a refractory period (which was the first possible explanation we considered).

*10) Fano Factor. Related to the last point: is it sensible to calculate Fano Factor (FF) for such a radically non-stationary process? Fano Factor is at best awkward for non-stationary processes and the main message of the paper is that spiking is non-stationary, e.g. extremely variable over large time scales and extremely precise when whiskers kinematics dictate so. For a Poisson process, FF = 1, independent of the window size. Since the observed FF for full stimulus window was large, evidently a Poisson process is not underlying spikes. At best it is a Poisson process driven by a variable mean rate. Indeed, Poisson statistics are meant to model stationary processes, that is, processes where the probability of an event per unit time is constant. Obviously firing probability increases in the tens of ms after touch, compared to the rest of the trial*.

We respectfully disagree. FF is appropriate for analysis of non-stationary processes. There are two types of non-stationarity that have different impact on the FF. First, a time-varying firing rate that is the same in every trial, referred to as an inhomogenous Poisson process. The Fano factor is often used to characterize non-stationary processes, precisely because FF = 1 even for an inhomogeneous Poisson process (e.g. Berry and Meister, 1997). Second, the other type of non-stationarity corresponds to changes in mean firing rate across trials. In this case, for a neuron that fires with Poisson statistics according to a stochastic rate in each trial, the Fano factor will be larger than 1 (i.e. it is a doubly stochastic process).

*11) Further, having demonstrated that the neurons of interest in barrel cortex do not resemble a Poisson process, the modeled neurons do:* “*We used a simplified model based on independent Poisson neurons with rapidly modulated spike rates*”*. It is puzzling to model Poisson neurons once one knows that the model is not correct*.

We now use spike trains sampled from recorded spike trains for modeling (new Figure 4).

*12) Factors accounting for spike trains. In the present analysis (in the second paragraph of the subsection “Low spike count variability”), 3 factors are used to account, one at a time, for spike trains: order of touch, whisker velocity, and whisker curvature. Why use them one at a time? Why not use all 3 at once and see how much better one does? Or use only the two main ones? Are there any spikes unaccounted for once one does this? This could be done with a GLM (perhaps enlisting the help of colleague Jeremy Freeman) or with similar approaches. Indeed, since the main message of the paper is that spike trains can be explained by whisker curvature and related kinematics, it would make sense to place less emphasis on the Poisson analysis and more emphasis on showing whisker state as a determining factor*.

The point of this figure is to show that touch-evoked spikes vary with the quality of the touch in multiple, correlated dimensions. This sets up the message of Figure 6, which is that spike count variability during touch is minimal once these touch characteristics are considered. Because of the low spike rates and spike counts, number of sensory variables, and limited experiment duration, it is difficult to fit models with more parameters.

The use of GLM models for this work is problematic for the following reasons:

A) Amount of data and overfitting. In a recent imaging study with a GLM-like model (Peron et al., 2015) we observed that we needed at least ∼100 trials to fit GLMs with only two variables and nine time bins (this was an imaging study with frame rate ∼ 7.5Hz). At a resolution of 1ms and much more temporal richness than in imaging we’d need many more time bins to describe the data, which increases the risk of overfitting. Regularization and carefully chosen temporal basis sets may overcome this problem under some conditions, but this is far from trivial. In general we are limited by trials and spikes.

B) GLMs typically use linear kernels and a static non-linearity. However, phase preference extracted from whisking cannot be linearly mapped. This means that standard GLMs cannot be applied to extract modulation with whisking.

C) Adaptation, to a first-order, can be accounted using a post-spike filter. But this significantly increases the number of model parameters and given limited data makes fitting impossible.

D) It is not possible in the classical GLM to generate near-binomial responses and still capture the overall ISI distribution.

E) Ideally every input variable should have a white-noise spectrum to fully sample the system. Temporally and spatially inhomogeneous sampling (across different input variables), as is the case in active behavior, can give unreliable estimates of the GLM parameters.

F) Adaptation of GLMs for trial-based, spiking neurons, with multiple correlated features were only recently developed (Park…Pillow, 2014). Even in this case, most of the inputs are discretized (saccade direction, stimulus onset, coherence level), implying relatively low dimensionality and few parameters.

This is just to say that we simply don’t know how to apply ‘GLM or with similar approaches’ (even with Dr. Freeman). To show that we have seriously considered this approach we show GLMs to fit the time-course of touch-responses (see Figure 7).

Author response image 1.GLM modeling of L4 touch responses. (a-b) Two example neural responses (blue) aligned to touch onset and their corresponding GLM prediction (red). (c-d) Same as (a-b) but aligned to touch offset. (e-f) Touch adaptation in function of touch number. (g-h) Proportion of spikes explained as a function of the exploration time (similar to Figure 3 in the paper). The blue curve (data) was obtained as described in the paper. For the GLM we started counting spikes ranked by the maximum GLM prediction (i.e. the moments with highest probability of a spike occurring according to the GLM model).**DOI:**
http://dx.doi.org/10.7554/eLife.06619.019

The GLM models touch onset and touch offset. To deal with adaptation, first touch and subsequent touches were treated separately. As expected, the GLM captures the touch-aligned responses well (constraints for smoothness mean that the GLM doesn’t capture sharp temporal responses). We tested if the GLM can help predict a larger fraction of spikes than the approach presented in the paper (Figure 3). Indeed, the GLM does slightly better (i.e. the proportion explained curves rise faster than with the approach we adapted) but the difference was small. We therefore chose to stick with the approach that was more closely related to the primary data.

*13) Other layers. Rather than focusing solely on L4, it would have been much more interesting to compare L4 with other cortical layers, and investigate the sources of variability there. Indeed, a major missing component is that neurons outside L4 are not recorded or reported. This is set up in the Introduction, but not really addressed. It's great to show L4 does one thing, but more interesting to show that it does something different than other layers. Given other publications from this group, it seems likely that the authors have other data that addresses this. Analyzing these data would make the paper much stronger*.

L4 is the core of this study. As discussed in point 5), other layers receive multiple long-range inputs that can make the activity more variable (since the input is not observed). However, we agree with the reviewers that adding L5 data would strength our paper by showing that the L4 is distinct from other layers. We repeated several of our analysis with a new set of L5 neurons. Whereas the vast majority of spikes in L4 can be explained by touch and whisking, only a small percentage of spikes can be explained in L5.

The new L5 data has been included in Figure 2 and Figure 3 and Figure 3—figure supplement 2.